# Investigating the Relationship between Climate and Hospital Admissions for Respiratory Diseases before and during the COVID-19 Pandemic in Brazil

Eduardo L. Krüger [1,*] and Anderson Spohr Nedel [2]

1 Civil Construction Department, Federal University of Technology, Curitiba Campus (Ecoville), Rua Deputado Heitor Alencar Furtado, 4900, Curitiba 81280-340, Brazil
2 Faculty of Agronomy, Federal University of Southern Border, Cerro Largo Campus, Cerro Largo 97900-000, Brazil
* Correspondence: ekruger@utfpr.edu.br

**Abstract:** This study aimed to analyze meteorological variables versus hospital admissions for respiratory diseases for the age groups of children under five and the elderly (over 65 years of age) in different climate regions of Brazil, for the years 2018 and 2020, i.e., before and after the outbreak of the COVID-19 pandemic. The aim of the study was, on one hand, to evaluate the influence of weather variables on respiratory disease in humans, and on the other hand, take into account two years with differing dynamics due to a worldwide pandemic that significantly changed people's lifestyles. The human biometeorological index (Universal Thermal Climate Index, UTCI) was used as representative of the integral association of meteorological variables. UTCI data were obtained from the ERA5-HEAT reanalysis database, which provides hourly grid data with a spatial resolution of 0.25° × 0.25°. The Brazilian cities Manaus, Brasilia and Porto Alegre, which represent different climatic contexts in the country, were used in the analysis. The method compared temperature and climate reanalysis data to hospital admission data for respiratory diseases, obtained from the Brazilian Unified Health System database (DATASUS), according to the International Classification of Diseases, Tenth Edition (ICD-10). Correlation analysis was performed between variables (hospital admissions versus climate-related data) in order to identify associations between them, also accounting for different time-lag effects. We analyzed seasonal influences on the obtained correlations, correlation strength and direct or inverse relationships. Results showed that the pandemic interfered in the association between morbidity due to respiratory illnesses and climate-related variables.

**Keywords:** morbidity; climate change; COVID 19; UTCI; hospital admissions

## 1. Introduction

One of the main issues discussed in the human biometeorology research field is climate variability, associated or not with urbanization and adverse effects of urban heat islands and extreme events. The most striking and often cited extreme event was the series of heat waves that struck Western Europe in 2003 and clearly showed how closely health risks are related with temperature changes [1]. The consequences of the 2003 heat wave were aggravated, as measured thermal conditions at the time were outside the range of expected climate variability.

In Brazil, heat waves have been reported to have become more frequent. In one of the most socially and economically vulnerable regions of the country, the northeast, a recent systematic evaluation of climatic trends for the time frame 1961–2014 showed an unequivocal increase in nighttime temperatures and in the frequency of heat waves accompanied by a drop in precipitation rates [2]. The latter can further exacerbate the ongoing desertification process in the inland parts of the Brazilian northeast. Increases in heat and decreases in precipitation, as direct impacts from climate change, have been

reported to be responsible for an amalgam of heat-related diseases and vector-borne diseases triggered by a rise in temperature [3–5]. Heat-related mortality is known to be most pronounced among the elderly [3].

Cold spells should not yet be disregarded as potential threats to human health, as unexpected cold fronts in a warming climate may be more health-threatening than the absolute temperature level itself [6]. The augmented health risk in this case might be associated with long-term acclimatization to a warming climate with corresponding changes in blood pressure having to come to terms with very short-term changes in temperature, along with a lack of behavioral response [7].

Vulnerable populations are likely to experience more severe impacts from climate variability above normal. Mortality and morbidity among the elderly due to climatic parameters were thoroughly presented in a systematic review and meta-analysis by Bunker et al. [8]. The study was focused in time-series and case-crossover studies and results showed that a similar 1 °C temperature increase or reduction leads to increases in health risks. The greatest risks were found to be associated with cold-induced pneumonia and respiratory morbidity and are expected to increase with climate change and global aging [8]. In a comparison among different all-cause excess deaths arising from the 2003 heat wave in France, Fouillet et al. [4] listed respiratory system diseases as the fourth most prevailing cause of death among the elderly (above 75 years of age), after heat-related illnesses and syncopes such as hyperthermia, dehydration and heat stroke, and just after circulatory diseases.

As for the very young, children under 5 years of age, Xu et al. [9] conducted a broad literature review on the interactions between climate and children's health and proposed that future research should focus on a number of aspects including socioeconomic status, children's vulnerability in low-income countries, mitigation and adaptation measures. Due to their physiology, metabolism, and behavior, children are more prone to suffer from climate change, particularly those under the age of 5 [10]. According to the World Health Organization, approximately one in five deaths worldwide occurs in a child of less than 5 years of age [11], and respiratory diseases, diarrhea, and malaria are responsible for over 50% of children's deaths [12]. Among children, lower respiratory tract infections are symptoms of the second largest disease with an environmental component, in global terms [12].

In their critical review, Kovats and Rajat [3] describe the most common methods to evaluate heat-related mortality and climatic data, time-series regression with the assessment of short-term associations between health data and exposure using multiple lag times and case-crossover studies, an approach that is a special case of the general case-series approach.

Using the time-series approach, a number of studies conducted in Brazil have looked into the relationship between climate and health data available in hospitals or in online platforms that provide data from the Brazilian health care system (relevant health-related data can be retrieved, e.g., from the Brazilian Unified Health System database—DATASUS). Costa et al. [13] tried to identify patterns of relationship between extreme weather conditions and hospitalization entries for cardiovascular diseases, obtained from the statistics sector at the University Hospital of Santa Maria in southern Brazil, using a time-series between 2012 and 2017. Souza and colleagues evaluated asthma-related hospitalization against climatic variables, pollutants and aerosols over time in Campo Grande, Mato Grosso do Sul [14]. Ikefuti et al. [15] used a similar method of regression analysis (Poisson regression model) to identify relationships between stroke mortality and air temperature in São Paulo using statistical data from the municipality. In southern Brazil and focusing in respiratory diseases, two studies analyzed asthma and bronchitis consultations and hospital admissions for pneumonia in children, respectively, against meteorological variables [16,17].

A recent systematic review conducted on studies published in the last decades in Brazil in human biometeorology identified gaps in knowledge [18] that should be filled. The systematic review was based on an extensive literature search in Scopus, Web of Science and Science Direct from the late 1990s until July 2021. Research gaps included the climate

dimensions of tourism, vector-borne diseases, mortality and morbidity data in urban areas. A follow-up symposium jointly organized by the authors of that paper showcased diverse studies conducted in the area of human biometeorology in Brazil [19]. On the interactions between health and climate, some of the authors of the above-mentioned studies were duly invited as speakers.

The COVID-19 pandemic, as of November 21, 2022, has caused an estimated total of 7.4 million deaths worldwide (IHME—https://www.healthdata.org/, accessed on 19 December 2022). With 700,000 deaths, Brazil accounts for about 10% of the estimated deaths attributable to the COVID-19 virus. During the first waves of the pandemic in 2020, still without any foreseeable vaccines, Brazil quickly moved up in the ranking of countries with the most cases and deaths related to the virus. The first wave hit the country through winter 2020 (from March to November), and lead to a general health crisis, as many hospitals were not prepared for receiving COVID-19 patients. In a comparison between Brazil and global COVID-19 data, the country was able to offer at the peak 8–13 beds/100 k patients during the first wave, while worldwide, that number was around 25/100 k. In addition, the federal government failed to provide institutional guidance and to take consistent actions against the spread and containment of the pandemic. Within Brazil, a few of the issues raised by Sachs et al. [20] became evident, such as the lack of coordination between central and local governments regarding suppression strategies. Amidst all this, patients with chronic respiratory diseases had limited access to hospitals and less chance to be admitted. In this context, it is likely that those patients received limited treatment during the pandemic.

The aim of the present study is to identify correlations between hospital admissions for respiratory diseases and climatic data for three different locations in Brazil, for two vulnerable population groups, children under 5 years of age and the elderly (above 65 years of age). In doing so, we also compare the relationship between climatic and hospitalization data before and after the outbreak of the COVID-19 pandemic. On one hand, we evaluate the influence of weather variables on respiratory disease in humans, while the other, we take into account two years with different dynamics due to a worldwide pandemic that significantly changed people's lifestyles. The novelty of this study is that such relationships have not yet been explored for Brazilian cities when accounting for the pandemic in 2020.

## 2. Methods

Three Brazilian capitals were analyzed, in order to evaluate in which way local climate features affect hospital admissions due to respiratory diseases. The three cities, Manaus, Brasília, and Porto Alegre, are located at different latitudes and longitudes, elevations, and proximity to the sea, and present very distinct climates as well as Köppen–Geiger classes (Table 1).

**Table 1.** Overall features of the cities evaluated.

| Location | Latitude in Degrees | Longitude in Degrees | Elevation (m a.s.l.) | Köppen–Geiger Class |
|---|---|---|---|---|
| Manaus | 3° S | 60° W | 39 | Am—monsoon climate |
| Brasília | 16° S | 48° W | 1130 | Aw—tropical with dry winters and rainy summer |
| Porto Alegre | 30° S | 51° W | 22 | Cfa—humid subtropical climate with hot summers |

### 2.1. Temperature and Climate Reanalysis Data

Hourly air temperature data were retrieved from the Brazilian institute of meteorology (INMET), and reanalysis data were extracted from the ERA5-HEAT dataset [21] in order to provide heat stress information in terms of the Universal Thermal Climate Index (UTCI, [22]). The multi-node, outdoor thermal comfort index UTCI has been used in a number of studies on heat stress in urban areas across the world, and has been found

suitable to Brazilian climates [23]. ERA5-HEAT reanalysis data proved to be meaningful for urban-related studies [24].

The study has a two-fold aim, namely: (1) to evaluate whether climate features (air temperature as a very basic independent variable, and the integrated index that represents the combination of air temperature and humidity, wind speed and the mean radiant temperature) have a verifiable impact on hospital admissions, and (2) to verify if the COVID-19 pandemic has interfered somehow in that relationship. In this context, two years served as benchmarks for analysis: the year 2018 and the first year of the pandemic, 2020. Air temperature data refer to the relevant WMO station for each location, which pertain to the Brazilian network of official automated meteorological stations. Hourly gridded reanalysis UTCI datasets were retrieved at $0.25° \times 0.25°$ spatial resolution from ERA5-HEAT for both years by means of the Copernicus Climate Data Store [25]. The grid cells used in the analyses roughly correspond to each city location.

### 2.2. Hospital Admission Data

Respiratory morbidity (hospital admission) data were retrieved from the publicly available database Unified Health System—DATASUS (www.datasus.gov.br, accessed on 11 November 2022), with records of authorizations for hospital admissions (AHA), Brazilian Ministry of Health, consisting of hospitalized patients. Datasets comprised place of residence, sex, age, primary diagnostics that led to admission (disease), date of admission and date of discharge. In this study, we focused on diseases of the respiratory system, codes varying between J00–J99 according to Disease International Code, version 10 (DIC 10) [26].

### 2.3. Analysis Procedure

The analytical part of the study was conducted based on descriptive statistics for climate and health data and on Pearson correlations found between daily data series.

### 3. Results

Air temperature data as measured at the respective meteorological stations for Manaus (WMO code 82331), Brasília (83377), and Porto Alegre (83967) are represented for both years in the whiskers plot in Figure 1.

Very slight differences were found between years in terms of air temperature at first glance. For Manaus, 2018 ($27.7 \pm 2.9$ °C, n = 359) was slightly cooler than 2020 ($27.9 \pm 2.7$ °C, n = 364); in Brasilia, both years were very much alike (in 2018: $21.2 \pm 3.7$ °C, n = 362; in 2020: $21.3 \pm 3.9$ °C, n = 366), and in Porto Alegre, there was no change in the annual mean, with a slight rise in the standard deviation (in 2018: $20 \pm 5.7$ °C, n = 365; in 2020: $20 \pm 5.9$ °C, n = 366).

Detailed analysis (Table 2) of differences above and below given set-point temperatures shows that overall, 2018 had reduced heat and cold stress levels (except for Manaus), when compared to 2020, for the assumed baseline temperatures above 25 °C and below 18 °C, respectively. Normalized data are presented, accounting for eventual gaps in the data series.

**Table 2.** Heating degree days (HDD) and cooling degree days (CDD) for baseline temperatures of 18 °C and 25 °C, respectively, for the three locations in a comparison between 2018 and 2020.

| Location | 2018 | | 2020 | |
| --- | --- | --- | --- | --- |
| | HDD | CDD | HDD | CDD |
| Manaus | - | 24,347 | - | 26,329 |
| Brasilia | 2783 | 3048 | 3102 | 3875 |
| Porto Alegre | 12,948 | 4815 | 13,376 | 5717 |

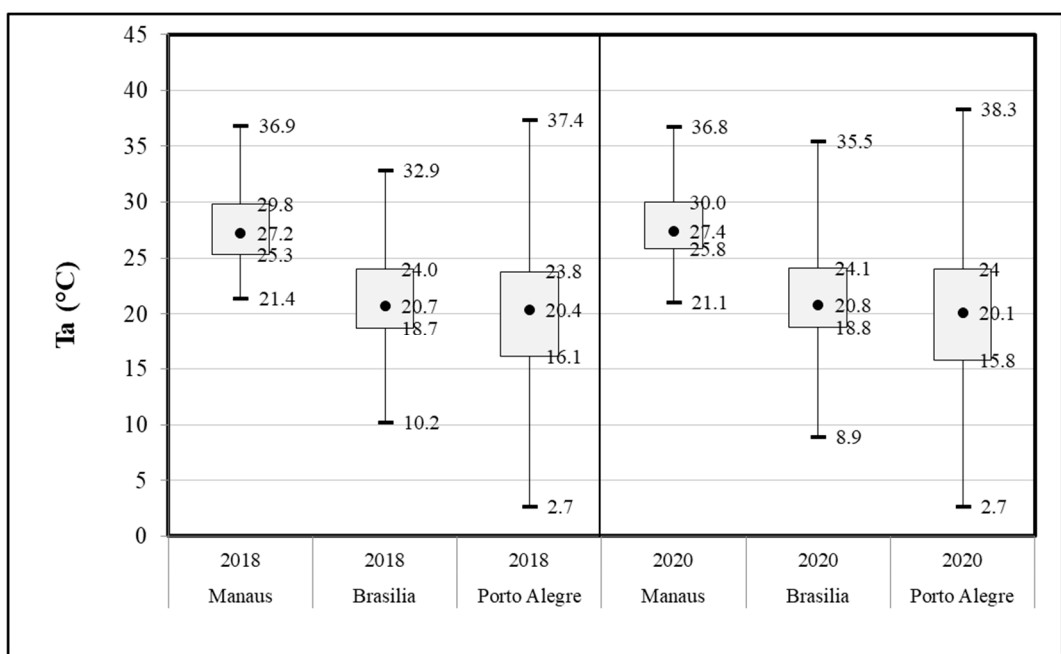

**Figure 1.** Boxplot with annual air temperatures (Ta) for the three locations in 2018 and 2020. Each box represents the first (Q₁ or 25th percentile) and third quartiles (Q₂ or 75th percentile), the whiskers represent the maximum and minimum values, and the central point represents the median.

The year 2020 had somewhat exacerbated HDD and CDD figures when compared to 2018, which suggests that over the year, fluctuations of the mean daily temperatures were higher in 2020.

In terms of thermal stress data (mean daily UTCI, Table 3), statistically significant changes, with $p < 0.01$, were observed for hourly values, albeit with very slight increases of up to 0.6 °C (UTCI scale). The distribution of thermal stress conditions in the three locations changed only slightly between years.

**Table 3.** Mean yearly UTCI and breakdown into UTCI thermal stress classes, for the three locations in a comparison between 2018 and 2020.

| UTCI Classes | Manaus | | Brasilia | | Porto Alegre | |
|---|---|---|---|---|---|---|
| | 2018 | 2020 | 2018 | 2020 | 2018 | 2020 |
| | % | % | % | % | % | % |
| slight cold stress | 0% | 0% | 0% | 0% | 7% | 9% |
| no thermal stress | 0% | 0% | 95% | 93% | 78% | 78% |
| moderate heat stress | 89% | 92% | 5% | 7% | 15% | 13% |
| strong heat stress | 11% | 8% | 0% | 0% | 0% | 0% |
| very strong heat stress | 0% | 0% | 0% | 0% | 0% | 0% |
| extreme heat stress | 0% | 0% | 0% | 0% | 0% | 0% |
| mean UTCI | 30.0 | 30.3 | 22.0 | 21.5 | 19.2 | 18.6 |

For the two years, hospital admissions for respiratory diseases for the two age groups varied significantly (Figure 2). In the three locations, numbers dropped (for children under 5, '<5 y-o' and the elderly, above 65 years of age '>65 y-o') for the availability of hospital beds due to the COVID-19 health crisis, and the risk of infection alongside a series of lockdowns imposed by the municipality drastically hindered healthcare service consultations in the public health system network.

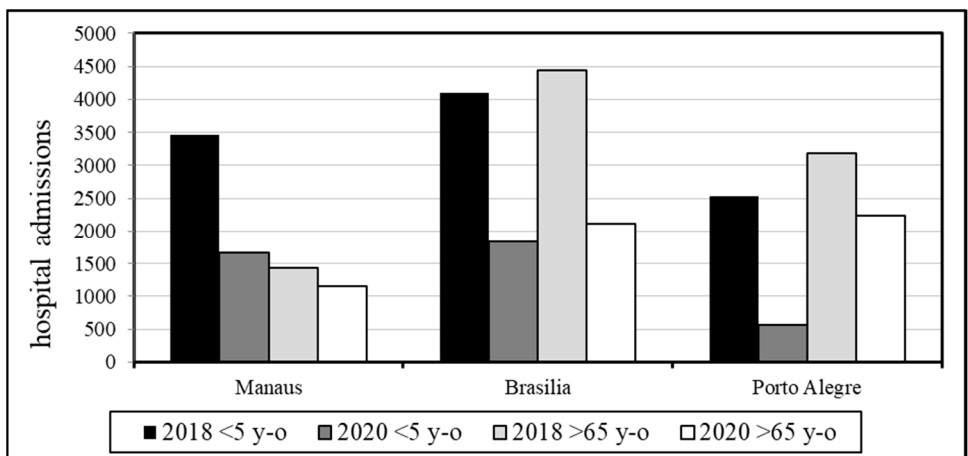

**Figure 2.** Total hospital admissions for children aged up to 5 years and the elderly, for the three locations in a comparison between 2018 and 2020.

The seasonal patterns of hospital admissions became evident when 2018 data were analyzed separately for the two age groups (Figure 3). In Porto Alegre, due to a higher fluctuation of thermal conditions over the year and well-defined seasons, and in Brasilia and Manaus, due to the dry-wet seasonality of those locations, these patterns were also seriously affected by wildfires in the north and central-west regions of the country, with all-cause hospital admissions attributable to wildfire-related pollution [27].

During the COVID-19 outbreak and particularly after the start of the spread of the virus in Brazil, by the end of February or mid-March, the seasonal pattern either disappeared entirely (as in the case of the children), or it was less discernible (the elderly) (Figure 4). The initial waves of the pandemic contributed to this confounding effect. During that time, a law was implemented and put into force from mid-March onward according to which telemedicine or remote medical care was introduced so as to reduce in-person consultations [28].

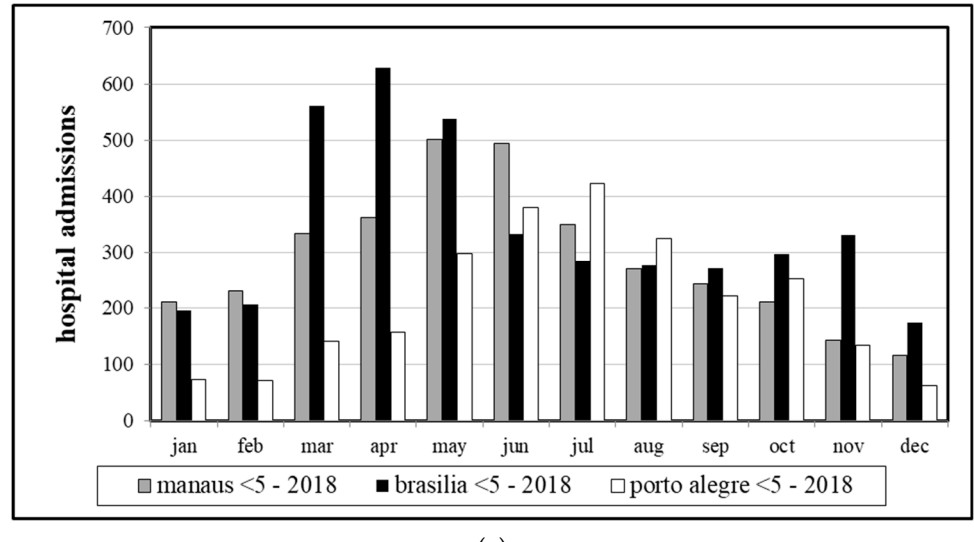

(a)

**Figure 3.** *Cont*.

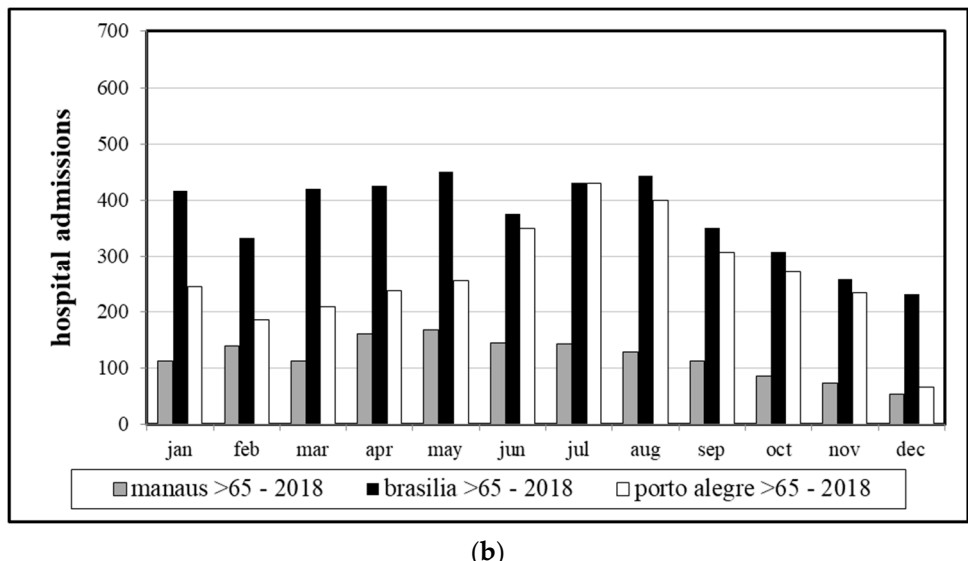

**(b)**

**Figure 3.** Total hospital admissions for children aged up to 5 years (**a**) and the elderly (**b**), for the three locations in a monthly comparison for 2018.

Correlations between hospital admissions and air temperature indicators (Ta, including: the 24 h mean, the nighttime mean, the daytime mean, the daily minimum and maximum temperature and the diurnal temperature range 'DTR', with running means up to six days ('lag6') before the date of admission 'lag0') and for thermal stress (daily mean UTCI, with lag times up to six days prior to admission) are discussed in detail. The lag or delay was used in similar analyses due to the delay in human body response to factors that triggered symptoms, resulting in a delay between the causing "event" and the search for medical care [17,29]. Table 4 shows aggregated data for Ta-related indicators, for children up to 5 years of age; Table 5 shows results for the elderly group.

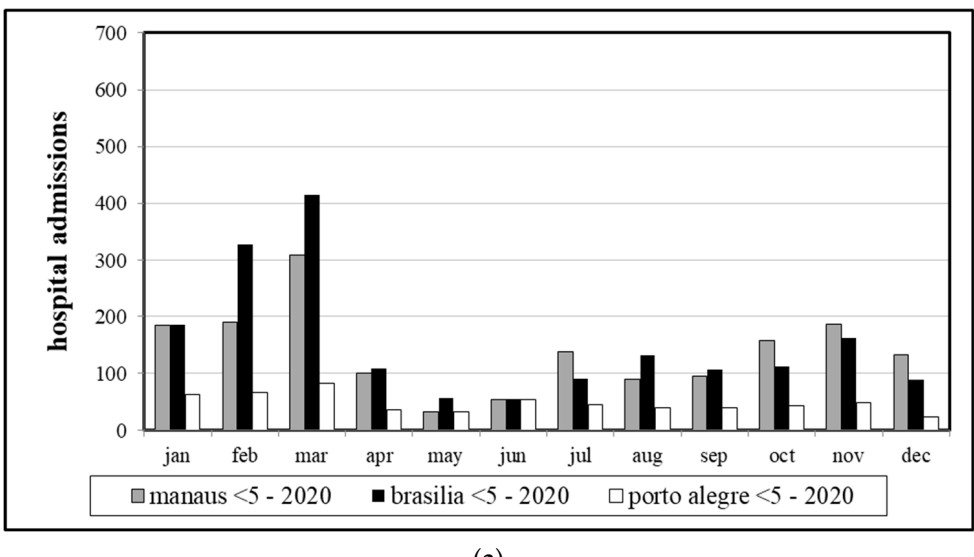

**(a)**

**Figure 4.** *Cont.*

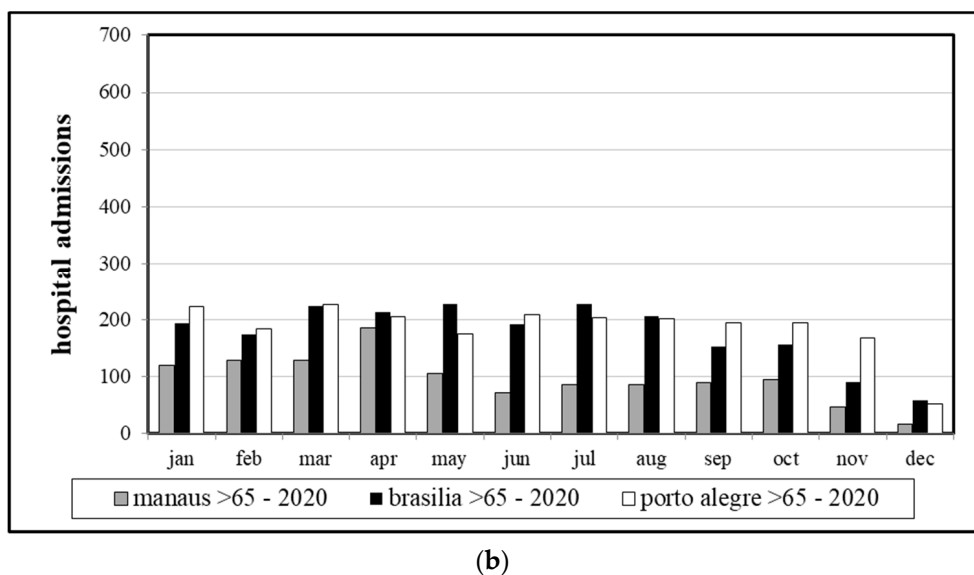

(**b**)

**Figure 4.** Total hospital admissions for children aged up to 5 years (**a**) and the elderly (**b**), for the three locations in a monthly comparison for 2020.

**Table 4.** Pearson correlations (r value) for hospital admissions for respiratory diseases versus Ta-related indicators for children aged up to 5 years, for the three locations on annual basis.

| Manaus | 2018 | | | | | | 2020 | | | | | |
|---|---|---|---|---|---|---|---|---|---|---|---|---|
| <5 y-o | 24 h Mean | Night Mean | Day Mean | Min | Max | DTR | 24 h Mean | Night Mean | Day Mean | Min | Max | DTR |
| lag0 | −0.13 | −0.13 | −0.12 | −0.16 | −0.17 | −0.11 | 0.00 | 0.08 | −0.05 | 0.10 | −0.02 | −0.11 |
| lag1 | −0.16 | −0.16 | −0.15 | −0.19 | −0.20 | −0.14 | 0.03 | 0.11 | −0.01 | 0.16 | 0.01 | −0.12 |
| lag2 | −0.18 | −0.17 | −0.18 | −0.22 | −0.24 | −0.18 | 0.04 | 0.12 | 0.00 | 0.17 | 0.02 | −0.11 |
| lag3 | −0.19 | −0.18 | −0.19 | −0.23 | −0.24 | −0.17 | 0.03 | 0.11 | −0.02 | 0.17 | 0.00 | −0.13 |
| lag4 | −0.20 | −0.19 | −0.20 | −0.25 | −0.25 | −0.19 | 0.04 | 0.12 | −0.02 | 0.19 | 0.00 | −0.15 |
| lag5 | −0.21 | −0.20 | −0.21 | −0.26 | −0.26 | −0.19 | 0.05 | 0.13 | 0.00 | 0.22 | 0.01 | −0.15 |
| lag6 | −0.21 | −0.21 | −0.21 | −0.28 | −0.26 | −0.18 | 0.05 | 0.13 | 0.00 | 0.22 | 0.01 | −0.15 |
| **Brasilia** | 2018 | | | | | | 2020 | | | | | |
| <5 y-o | 24 h Mean | Night Mean | Day Mean | Min | Max | DTR | 24 h Mean | Night Mean | Day Mean | Min | Max | DTR |
| lag0 | −0.13 | −0.08 | −0.17 | 0.00 | −0.17 | −0.15 | 0.19 | 0.26 | 0.13 | 0.36 | 0.07 | −0.25 |
| lag1 | −0.15 | −0.09 | −0.18 | 0.00 | −0.19 | −0.16 | 0.19 | 0.25 | 0.13 | 0.36 | 0.08 | −0.26 |
| lag2 | −0.15 | −0.09 | −0.19 | 0.00 | −0.21 | −0.18 | 0.19 | 0.25 | 0.13 | 0.36 | 0.07 | −0.27 |
| lag3 | −0.15 | −0.09 | −0.18 | 0.01 | −0.20 | −0.18 | 0.18 | 0.25 | 0.12 | 0.36 | 0.06 | −0.29 |
| lag4 | −0.14 | −0.08 | −0.18 | 0.01 | −0.20 | −0.19 | 0.18 | 0.25 | 0.12 | 0.36 | 0.05 | −0.30 |
| lag5 | −0.14 | −0.07 | −0.19 | 0.01 | −0.21 | −0.19 | 0.18 | 0.25 | 0.12 | 0.37 | 0.05 | −0.32 |
| lag6 | −0.14 | −0.07 | −0.19 | 0.02 | −0.21 | −0.20 | 0.18 | 0.25 | 0.11 | 0.37 | 0.05 | −0.33 |
| **Porto Alegre** | 2018 | | | | | | 2020 | | | | | |
| <5 y-o | 24 h Mean | Night Mean | Day Mean | Min | Max | DTR | 24 h Mean | Night Mean | Day Mean | Min | Max | DTR |
| lag0 | −0.67 | −0.66 | −0.66 | −0.64 | −0.61 | −0.67 | 0.11 | 0.11 | 0.11 | 0.09 | 0.11 | 0.07 |
| lag1 | −0.69 | −0.68 | −0.69 | −0.67 | −0.65 | −0.69 | 0.12 | 0.13 | 0.12 | 0.11 | 0.12 | 0.05 |
| lag2 | −0.71 | −0.70 | −0.71 | −0.69 | −0.67 | −0.71 | 0.15 | 0.14 | 0.15 | 0.13 | 0.15 | 0.08 |
| lag3 | −0.72 | −0.71 | −0.72 | −0.70 | −0.69 | −0.72 | 0.17 | 0.16 | 0.17 | 0.14 | 0.18 | 0.12 |
| lag4 | −0.73 | −0.72 | −0.73 | −0.71 | −0.71 | −0.73 | 0.18 | 0.17 | 0.18 | 0.15 | 0.20 | 0.15 |
| lag5 | −0.74 | −0.73 | −0.74 | −0.72 | −0.72 | −0.74 | 0.19 | 0.18 | 0.19 | 0.16 | 0.21 | 0.18 |
| lag6 | −0.74 | −0.73 | −0.74 | −0.72 | −0.73 | −0.74 | 0.19 | 0.18 | 0.20 | 0.17 | 0.22 | 0.18 |

**Table 5.** Pearson correlations (r value) for hospital admissions for respiratory diseases versus Ta-related indicators for the age group above 65 years, for the three locations on annual basis.

| Manaus | | | | 2018 | | | | | | 2020 | | |
|---|---|---|---|---|---|---|---|---|---|---|---|---|
| >65 y-o | 24 h Mean | Night Mean | Day Mean | Min | Max | DTR | 24 h Mean | Night Mean | Day Mean | Min | Max | DTR |
| lag0 | −0.07 | −0.07 | −0.06 | −0.12 | −0.07 | 0.00 | 0.09 | 0.11 | 0.07 | 0.07 | 0.05 | 0.01 |
| lag1 | −0.07 | −0.07 | −0.07 | −0.13 | −0.09 | −0.02 | 0.08 | 0.10 | 0.06 | 0.07 | 0.04 | 0.00 |
| lag2 | −0.08 | −0.09 | −0.08 | −0.14 | −0.10 | −0.02 | 0.09 | 0.12 | 0.07 | 0.09 | 0.06 | 0.00 |
| lag3 | −0.10 | −0.11 | −0.08 | −0.16 | −0.10 | −0.01 | 0.12 | 0.15 | 0.10 | 0.12 | 0.06 | −0.01 |
| lag4 | −0.11 | −0.12 | −0.09 | −0.16 | −0.11 | −0.02 | 0.12 | 0.16 | 0.10 | 0.13 | 0.06 | −0.02 |
| lag5 | −0.12 | −0.12 | −0.10 | −0.17 | −0.11 | −0.02 | 0.15 | 0.15 | 0.09 | 0.13 | 0.05 | −0.03 |
| lag6 | −0.13 | −0.14 | −0.12 | −0.18 | −0.13 | −0.04 | 0.12 | 0.16 | 0.09 | 0.15 | 0.05 | −0.04 |
| Brasilia | | | | 2018 | | | | | | 2020 | | |
| >65 y-o | 24 h Mean | Night Mean | Day Mean | Min | Max | DTR | 24 h Mean | Night Mean | Day Mean | Min | Max | DTR |
| lag0 | −0.15 | −0.15 | −0.12 | −0.20 | −0.08 | 0.10 | −0.13 | −0.15 | −0.10 | −0.16 | −0.09 | 0.06 |
| lag1 | −0.16 | −0.17 | −0.13 | −0.22 | −0.10 | 0.11 | −0.13 | −0.16 | −0.10 | −0.17 | −0.09 | 0.07 |
| lag2 | −0.17 | −0.18 | −0.15 | −0.23 | −0.11 | 0.11 | −0.16 | −0.19 | −0.13 | −0.20 | −0.11 | 0.08 |
| lag3 | −0.18 | −0.19 | −0.16 | −0.23 | −0.11 | 0.12 | −0.19 | −0.21 | −0.16 | −0.21 | −0.14 | 0.07 |
| lag4 | −0.19 | −0.19 | −0.17 | −0.24 | −0.12 | 0.12 | −0.21 | −0.22 | −0.19 | −0.22 | −0.17 | 0.05 |
| lag5 | −0.19 | −0.19 | −0.18 | −0.24 | −0.12 | 0.12 | −0.23 | −0.23 | −0.20 | −0.23 | −0.18 | 0.05 |
| lag6 | −0.19 | −0.19 | −0.18 | −0.24 | −0.13 | 0.12 | −0.22 | −0.23 | −0.21 | −0.22 | −0.19 | 0.03 |
| Porto Alegre | | | | 2018 | | | | | | 2020 | | |
| >65 y-o | 24 h Mean | Night Mean | Day Mean | Min | Max | DTR | 24 h Mean | Night Mean | Day Mean | Min | Max | DTR |
| lag0 | −0.43 | −0.44 | −0.42 | −0.42 | −0.38 | −0.07 | −0.01 | −0.01 | −0.02 | −0.03 | 0.00 | 0.04 |
| lag1 | −0.46 | −0.47 | −0.45 | −0.46 | −0.42 | −0.07 | −0.02 | −0.02 | −0.02 | −0.05 | 0.00 | 0.07 |
| lag2 | −0.50 | −0.50 | −0.49 | −0.49 | −0.46 | −0.08 | −0.03 | −0.03 | −0.02 | −0.06 | 0.01 | 0.14 |
| lag3 | −0.52 | −0.52 | −0.52 | −0.51 | −0.49 | −0.12 | −0.03 | −0.04 | −0.03 | −0.07 | 0.01 | 0.14 |
| lag4 | −0.54 | −0.53 | −0.53 | −0.53 | −0.51 | −0.14 | −0.04 | −0.05 | −0.03 | −0.07 | 0.00 | 0.15 |
| lag5 | −0.55 | −0.55 | −0.54 | −0.54 | −0.53 | −0.15 | −0.06 | −0.06 | −0.05 | −0.08 | −0.02 | 0.14 |
| lag6 | −0.56 | −0.55 | −0.55 | −0.55 | −0.54 | −0.16 | −0.07 | −0.07 | −0.06 | −0.09 | −0.04 | 0.12 |

Overall, the strength of correlations between variables dropped in 2020, which suggests that hospital admissions due to respiratory diseases in that year were not as clearly associated as before the pandemic with climate-related factors. All correlations were found to be statistically significant, with $p < 0.01$ (obtained from $t$ tests, two-tailed, 'type 3', with unequal variances). An exception was Brasilia, where hospital admissions had stronger correlations to Ta-related variables in the first year of the COVID-19 pandemic than in 2018, albeit correlation strength in both cases was weak at most.

The southernmost location, Porto Alegre, with defined seasons, had moderate and strong correlations for 2018 for the two age groups, respectively, which completely vanished to negligible/weak correlations in 2020. In 2018, the inverse correlations showed that low temperatures and low DTR (which could be related to cold, overcast days with a low diurnal temperature swing) were associated with a rise in hospital admissions. That effect consistently rose for increasing lag time prior to admission. For the children group, there was even an inversion of that relationship in 2020, even though the lag effect persisted.

For Manaus, with year-long temperature profiles unchanged and no seasonal temperature profile (cf. Figure 1), correlations with Ta-related variables were much lower than in Porto Alegre, with either negligible or weak correlations, which in 2020 even showed an inverse relationship. Figure 5 exemplifies graphically the loss of correlation strength and the inversion of its value for the children group and, less intensively for the elderly group, for Porto Alegre, at lag6.

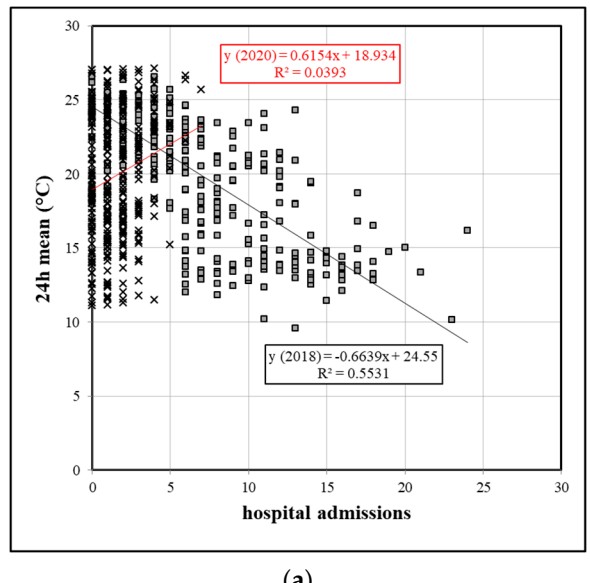

(**a**)

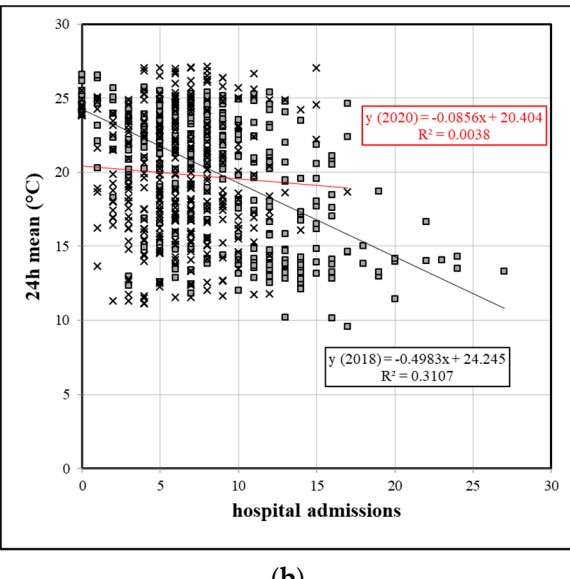

(**b**)

**Figure 5.** Scatter plots for hospital admissions versus 24 h mean Ta, for the children (**a**) and the age group above 65 years (**b**) in Porto Alegre.

Tables 6 and 7 show correlations found between hospital admissions and mean daily UTCI data.

**Table 6.** Pearson correlations (r value) for hospital admissions for respiratory diseases versus mean daily UTCI for children aged up to 5 years, for the three locations on annual basis.

|  | **Manaus** | | **Brasilia** | | **Porto Alegre** | |
|---|---|---|---|---|---|---|
| **<5 y-o** | **2018** | **2020** | **2018** | **2020** | **2018** | **2020** |
| lag0 | −0.19 | 0.17 | −0.12 | 0.31 | −0.62 | 0.10 |
| lag1 | −0.22 | 0.23 | −0.13 | 0.32 | −0.65 | 0.13 |
| lag2 | −0.23 | 0.24 | −0.13 | 0.33 | −0.67 | 0.14 |
| lag3 | −0.24 | 0.24 | −0.12 | 0.34 | −0.69 | 0.16 |
| lag4 | −0.25 | 0.25 | −0.11 | 0.35 | −0.70 | 0.17 |
| lag5 | −0.26 | 0.28 | −0.10 | 0.36 | −0.71 | 0.18 |
| lag6 | −0.26 | 0.29 | −0.10 | 0.37 | −0.71 | 0.18 |

**Table 7.** Pearson correlations (r value) for hospital admissions for respiratory diseases versus mean daily UTCI for the age group above 65 years, for the three locations on annual basis.

|  | **Manaus** | | **Brasilia** | | **Porto Alegre** | |
|---|---|---|---|---|---|---|
| **>65 y-o** | **2018** | **2020** | **2018** | **2020** | **2018** | **2020** |
| lag0 | −0.15 | −0.08 | −0.18 | −0.18 | −0.43 | −0.05 |
| lag1 | −0.16 | −0.10 | −0.21 | −0.18 | −0.46 | −0.04 |
| lag2 | −0.18 | −0.13 | −0.23 | −0.21 | −0.50 | −0.04 |
| lag3 | −0.19 | −0.13 | −0.24 | −0.24 | −0.53 | −0.04 |
| lag4 | −0.19 | −0.14 | −0.25 | −0.25 | −0.55 | −0.04 |
| lag5 | −0.20 | −0.15 | −0.26 | −0.26 | −0.55 | −0.06 |
| lag6 | −0.20 | −0.15 | −0.26 | −0.26 | −0.56 | −0.07 |

A similar pattern was found for UTCI correlations as for Ta indicators. An inversion of negative to positive correlations between years for the three locations, with the strongest correlations for the southernmost location in 2018, with significantly lower correlations in 2020, was noticed in the children group. The lag effect proved to be meaningful in all datasets, with correlations becoming stronger the longer the lag was, that is, the response to a climate-related event took longer to result in hospital admissions. For the elderly group, all correlations were negative, losing their strength in 2020. In Brasilia, the year of the pandemic did not alter the pattern of relationships for that age group.

## 4. Discussion

Although correlation analysis does not imply a causal relationship between climate-related variables and hospital admissions, it is a useful procedure to test whether an association between different variables exists. In the case of hospital admissions due to respiratory diseases, results suggest that seasonal parameters do affect that relationship and that the strength of such an association is stronger in well-defined climates and less so in tropical regions with unvaried thermal conditions over the year. The existence of wet and dry seasons with related wildfires also contributes to the association between hospital admissions and local climate features (as in Brasilia). The correlation analysis of the same climate-related parameters with hospital admissions during the COVID-19 pandemic showed an impact of the pandemic on that relationship for the two age groups in the three locations studied.

As both years did not show relevant changes in climate-related (Ta and UTCI) data, it can be assumed that complications in healthcare during the pandemic were largely responsible for the change in the patterns of relationship, as verified in this study. The fact that significantly fewer patients were admitted to hospitals due to respiratory diseases during 2020 is emblematic and has been observed in other parts of the world as well [30–32]. In Brazil, several factors contributed to this decrease, and some of the most critical factors were the healthcare system's capacity, the lack of available hospital beds, and the higher risk of infections due to the COVID-19 pandemic [33,34]. The burden on the Brazilian healthcare system during the pandemic was also heterogeneous, and certain regions of the country, such as Manaus, one of the analyzed cities, showed higher healthcare vulnerability than others [35]. In Manaus, the reduction in hospital admissions of patients with diseases other than COVID-19 was due to the shortage of available beds, ICUs, and respiratory equipment at the time, following a crisis in the funeral services sector during the first wave [36]. Further contributing to that was a lack of attention by the local population to restrictions imposed by the pandemic, such as social distancing, restricted mobility, etc. [36].

Thus, the relationship between climate-related variables was weakened and even inversed as COVID-19 patients gained priority during the first waves of the pandemic in 2020. Despite the strong temperature and climate dependency of respiratory diseases such as pneumonia, Porto Alegre, characterized by having well-defined seasons with several cold fronts during winter, showed a drastic decline in the association between meteorological variables and hospital admissions. Previous studies suggest that in the metropolitan region of Porto Alegre in the years before the pandemic, the relationship between meteorological conditions and outpatient consultations for asthma or bronchitis in children (below 9 years of age) led to negative correlations and that the highest frequencies of respiratory complications occurred in winter [16].

Indeed, several epidemiological studies have reported that short-term exposure to both cold and hot temperatures can be associated with increased hospital admissions from respiratory diseases, and a U-shaped curve usually defines such a relationship. Chai et al. [37], in a study using a distributed lag non-linear model coupled with a generalized additive model for estimating the association between air temperature and hospital outpatient visits for respiratory diseases, for a large sample of 1,042,656 hospital visits over 10 years, came up with relative risk curves for respiratory diseases for different age groups. In Lanzhou,

a semi-arid region in northwest China, the relative risk was found to be the highest for children between 6–14 years of age, but, in a comparison between children aged up to 5 years and the elderly (above 60 years of age), the risk increased for the latter [37].

The consideration of an outdoor thermal comfort index (the UTCI) in the analysis either improved the strength of the correlations or led to similar trends as observed in the Ta-related indicators. Figure 6 shows the comparison between the closest variable to the daily mean UTCI, the Ta 24 h mean (daily mean air temperature) and the UTCI daily mean, for lag6. It is surprising that the correlations follow a similar trend, as reanalysis data follow a grid that does not fully correspond to that of the WMO meteorological station, where field data came from. A previous work, however, suggests that UTCI reanalysis data are reasonable surrogates for missing field data in OTC studies involving questionnaire surveys [24]. Despite the fact that there were some discrepancies between Ta and UTCI data in terms of correlation strength (as in Manaus, for the children group in 2020, improving correlations when UTCI data are accounted for), for the two age groups at the location with marked seasons and with the strongest correlations in 2018 (Porto Alegre), there was no noticeable improvement in the correlations when UTCI data were used instead of the ambient temperature data. In general, though, using UTCI instead of Ta led to similar or even higher correlations with hospital admission data. Thus, UTCI reanalysis data can also serve as feasible surrogates of Ta in analyses involving morbidity due to respiratory diseases and estimated thermal stress data.

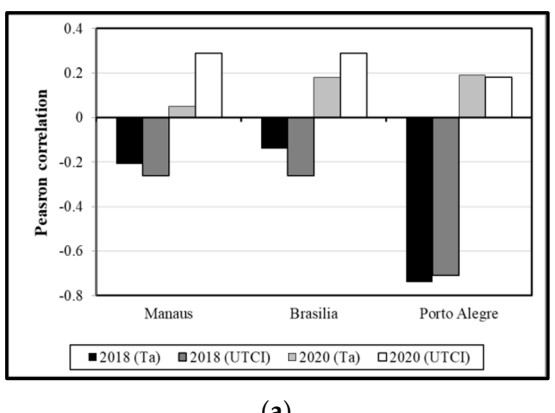 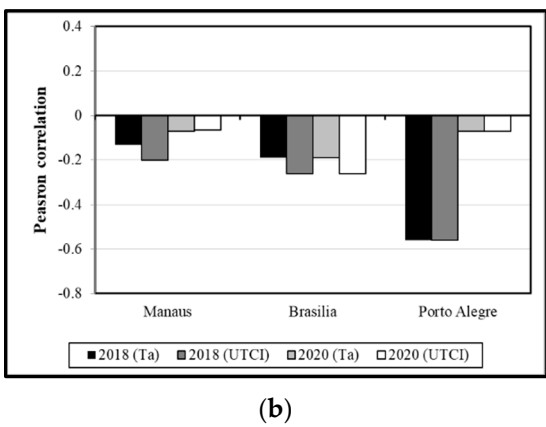

(**a**)　　　　　　　　　　　　　　　　　　　　　　　　　　(**b**)

**Figure 6.** Comparison of Pearson correlations (r value) for hospital admissions with respiratory diseases versus mean daily UTCI for the age group below 5 years (**a**) and above 65 years (**b**), for the three locations on annual basis—daily mean temperatures versus UTCI data (lag6).

## 5. Conclusions

The study looked at possible impacts of the COVID-19 pandemic on hospital admissions due to respiratory illnesses. The relationship between air temperature-related variables and mean daily UTCI on an annual basis showed that such an association was stronger in subtropical conditions, gradually losing its power in hot-humid tropical conditions. The pandemic confounded this relationship. The drop in hospital admissions due to prioritized treatment of COVID-19 cases, the weakness of the healthcare system and the restrictions imposed by the pandemic are suggested to have interfered in this process.

Patients with weather-related respiratory issues were on their own, received limited or self-prescribed medication, or passed away. The lack of preparedness of the healthcare system is part of the problem. Telemedicine with remote medical care remained in most cases a limited solution, as low-income populations and remote areas frequently lack internet access. This calls for improving the resiliency of the healthcare system for future pandemics, so that the system can absorb a large increase in patients in a high-demand situation. Another line of work is to improve microclimates and also indoor thermal conditions, as patients with respiratory diseases have likely stayed home during mobility

restrictions, social distancing and lockdown measures, thus having been less exposed to climatic conditions.

Some limitations of this exploratory work can be mentioned. The analysis was based on descriptive statistics and on correlation analysis, only. Advanced statistics such as additive models (AM), generalized additive models (GAM), or Poisson distributions, among other non-linear relationships, had not been investigated at this stage. The study was also limited to three cities with quite distinct socio-economic conditions, and with differences in their management of local health services during the pandemic.

Further work should scrutinize the metadata used for this analysis. As pointed out by Bunker et al. [8], effects of heat on morbidity and mortality due to respiratory diseases are most often immediate, while effects of cold become more evident with longer lag times. A sensitivity analysis would be advisable. Another interesting subject of analysis could be the intergroup differences between children and the elderly regarding health effects in the three locations, considering the fact that especially the elderly spend most of their time indoors. Finally, research initiatives should be started with a focus on the fate of patients with respiratory diseases when confronted with restrictions regarding hospital admissions during the pandemic.

Finally, it should be stressed that ethical concerns were not applicable in this research, as all relevant health data were freely available in anonymized format at the DATASUS website.

**Author Contributions:** Conceptualization, E.L.K. and A.S.N.; methodology, E.L.K. and A.S.N.; software, E.L.K. and A.S.N.; validation, E.L.K. and A.S.N.; formal analysis, E.L.K.; investigation, E.L.K. and A.S.N.; resources, E.L.K. and A.S.N.; data curation, E.L.K. and A.S.N.; writing—original draft preparation, E.L.K.; writing—review and editing, E.L.K.; visualization, E.L.K. and A.S.N.; supervision, E.L.K. and A.S.N.; project administration, E.L.K. and A.S.N.; funding acquisition, no funding. All authors have read and agreed to the published version of the manuscript.

**Funding:** This research received no external funding.

**Institutional Review Board Statement:** Not applicable.

**Informed Consent Statement:** Not applicable.

**Data Availability Statement:** Data can be made available under reasonable request.

**Conflicts of Interest:** The authors declare no conflict of interest.

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
