# Peer review of "Investigating the Relationship between Climate and Hospital Admissions for Respiratory Diseases before and during the COVID-19 Pandemic in Brazil"

_sustainability, doi:10.3390/su15010288_

Round 1

Reviewer 1 Report

The authors introduces a new work, the paper is interesting 

but some comments should be done before accepting

1- The abstract should be cover all aspects of the paper 

2- Figures are good , but som are not in place

3- The novelty should be clear 

4- The conclusion should contains all major findings 

5- The future work should be more specified 

after all I can accept the paper 

Author Response

Dear reviewer,

We have made slight changes in the manuscript so as to make it a little more precise:

1- The abstract should be cover all aspects of the paper - revised/ added text

2- Figures are good , but som are not in place - figures depend on the final format of the paper, which will be arranged by the editing team.

3- The novelty should be clear  - novelty has been pointed out at the end of the introduction

4- The conclusion should contains all major findings - I have read the conclusions and that part is, in my opinion, complete enough

5- The future work should be more specified - I have read the conclusions and that part is, in my opinion, complete enough

after all I can accept the paper 

Reviewer 2 Report

I would like to recommend this manuscript for publication after minor revision:

1.     The Keywords, "climate change" should be revised as "Climate change".

2.     The Introduction, it is too long and the paragraphs are too trivial. In scientific articles, the introduction usually does not exceed 5 paragraphs.

3.     The title, it should be noted that the research is aimed at parts of Brazil.

4.     Figure 1, the legend should be marked (mean ± SD, n=?), which indicate how many times the same group of data is repeatedly investigated or tested.

5.     The format of references is not according to the MDPI. Please revise it.

Author Response

Dear reviewer,

Our revised version has been prepared as follows:

1.     The Keywords, "climate change" should be revised as "Climate change". 
R: Done

2.     The Introduction, it is too long and the paragraphs are too trivial. In scientific articles, the introduction usually does not exceed 5 paragraphs.
R: The first and third paragraphs have been removed. The remaining paragraphs present useful information, necessary to the topic discussion. (reference ISB 2022 has been duly removed from the text)

3.     The title, it should be noted that the research is aimed at parts of Brazil.
R: Thanks, this is a good tip! The new title is as follows "Investigating the relationship between climate and hospital admissions for respiratory diseases before and during the COVID-19 pandemic in Brazil"

4.     Figure 1, the legend should be marked (mean ± SD, n=?), which indicate how many times the same group of data is repeatedly investigated or tested.
R: These are data for the two years on daily basis, though there were slight gaps in the series. The box plot has now numeric infrmation and a text was added below Figure 1.

5.     The format of references is not according to the MDPI. Please revise it.

R: Adjusted

Reviewer 3 Report

This study is interesting and very systematically established.

The aim of the present study is to identify correlations between hospital admissions 135 for respiratory diseases and climatic data for three different locations in Brazil, for two 136 vulnerable population groups, children under five years of age and the elderly (above 65 137 year-olds). It is also compared the relationship between climatic and hospitalization data before and after the outbreak of the COVID-19 pandemic.

Methodology is well described - Three Brazilian capitals were analyzed, in order to evaluate in which way local climate features affect hospital admissions due to respiratory diseases.

The analytical part of the study was conducted based on descriptive statistics for climate and health data and on Pearson correlations found between daily data series

In conclusions authors confirmed the  possible impacts of the COVID-19 pandemic on hospital admissions due to respiratory illnesses. The relationship between air temperature-related variables and mean daily UTCI on annual basis showed that such association is stronger in subtropical conditions, gradually losing its power in hot-humid tropical conditions. The pandemic confounded such relationship. The drop in hospital admissions due to prioritized treatment of COVID-19 cases, the weakness of the healthcare system and the several restrictions imposed by the pandemic are suggested to have interfered in this  process. 

Explore the limitations of this study are necessary. 

ethical concerns - suggestion to inform that ethical concerns are not necessary

Author Response

Dear reviewer,

Thanks for your input. Responses are below:

This study is interesting and very systematically established.
R: Thank you!

The aim of the present study is to identify correlations between hospital admissions 135 for respiratory diseases and climatic data for three different locations in Brazil, for two 136 vulnerable population groups, children under five years of age and the elderly (above 65 137 year-olds). It is also compared the relationship between climatic and hospitalization data before and after the outbreak of the COVID-19 pandemic.

Methodology is well described - Three Brazilian capitals were analyzed, in order to evaluate in which way local climate features affect hospital admissions due to respiratory diseases.

The analytical part of the study was conducted based on descriptive statistics for climate and health data and on Pearson correlations found between daily data series

In conclusions authors confirmed the  possible impacts of the COVID-19 pandemic on hospital admissions due to respiratory illnesses. The relationship between air temperature-related variables and mean daily UTCI on annual basis showed that such association is stronger in subtropical conditions, gradually losing its power in hot-humid tropical conditions. The pandemic confounded such relationship. The drop in hospital admissions due to prioritized treatment of COVID-19 cases, the weakness of the healthcare system and the several restrictions imposed by the pandemic are suggested to have interfered in this  process. 
R: The reviewer has captured the general idea,

Explore the limitations of this study are necessary. 
R: Some limitations have been listed.

ethical concerns - suggestion to inform that ethical concerns are not necessary
R: OK